# Traumatic dislocation of middle ear ossicles: A new computed tomography classification predicting hearing outcome

**Georgios Mantokoudis**[1]*, **Njima Schläpfer**[1], **Manuel Kellinghaus**[2], **Arsany Hakim**[2], **Moritz von Werdt**[1], **Marco D. Caversaccio**[1], **Franca Wagner**[2]

**1** Department of Otorhinolaryngology, Head and Neck Surgery, Inselspital, Bern University Hospital, University of Bern, Bern, Switzerland, **2** University Department of Diagnostic and Interventional Neuroradiology, Inselspital, Bern University Hospital, University of Bern, Bern, Switzerland

* georgios.mantokoudis@insel.ch

## Abstract

### Objectives

To assess the feasibility of radiologic measurements and find out whether hearing outcome could be predicted based on computer tomography (CT) scan evaluation in patients with temporal bone fractures and suspected ossicular joint dislocation.

### Methods

We assessed 4002 temporal bone CT scans and identified 34 patients with reported ossicular joint dislocation due to trauma. We excluded those with no proven traumatic ossicular dislocation in CT scan and patients with bilateral temporal bone fractures. We measured four parameters such as malleus-incus axis distance, malleus-incus angle at midpoints, malleus- incus axis angle and ossicular joint space. The contralateral healthy side served as its own control. Hearing outcome 1–3 months after the index visit was analyzed. We assessed diagnostic accuracy and performed a logistic regression using radiologic measurement parameters for outcome prediction of conductive hearing loss (defined as >20dB air-bone gap).

### Results

We found excellent inter-rater agreement on the measurement of axis deviation between incus and malleus in CT scans (interclass correlation coefficient 0.81). The larger the deviation of incus and malleus axis, the higher probability of poor hearing outcome (odds ratio (OR) 2.67 per 0.1mm, $p = .006$). A cut-off value for the axis deviation of 0.25mm showed a sensitivity of 0.778 and a specificity of 0.94 ($p < .001$) for discrimination between poor and good hearing outcome in terms of conductive hearing loss.

### Conclusion

Adequate assessment of high resolution CT scans of temporal bone in which ossicular chain dislocation had occurred after trauma was feasible. Axis deviations of the incus and

**Data Availability Statement:** The data underlying this study are available on the Bern Open Repository (https://boris.unibe.ch/147873/).

**Funding:** G.M. received funding from the Swiss National Science Foundation (#320030_173081). All other authors received no external financial support for that study. The funders had no role in study design, data collection and analysis, decision to publish, or preparation of the manuscript.

**Competing interests:** The authors have declared that no competing interests exist.

the malleus were strongly predictive for poor hearing outcome in terms of air conduction 1–3 months after trauma. We propose a 3-level classification system for hearing outcome prediction based on radiologic measures.

## Introduction

Ossicular chain dislocation is often associated with a traumatic fracture of the temporal bone [1–3]. Often hearing dysfunction is overlooked in polytrauma patients because other trauma-related physical/brain injuries take medical priority. Immediate hearing assessment and outcome prediction in cases with suspected hearing loss is not possible because patients often have a traumatic head or brain injury and are not eligible for hearing tests in the acute stage. Even if they do not require bed rest, patients might still not be able to participate in audiometry since they might have cognitive impairment. Finally, temporal bone fractures are often associated with a hemotympanum, which makes an accurate early assessment of the middle ear impossible. Currently, patients with a suspected traumatic dislocation of the ossicular chain only undergo a comprehensive hearing test after some weeks or months, and the risk for loss to follow-up is high. Unilateral hearing loss (conductive or sensorineural) might therefore remain untreated in a large proportion of patients [4, 5]. Any technique to predict hearing outcome at the initial assessment might help to initiate early follow-up treatment with hearing aids, cochlear implants, or reconstructive middle ear surgery. Temporal bone computed tomography (CT) assessing the length of the fracture line sparing the otic capsule was reported to be useful for predicting sensorineural hearing loss; however, no prediction was made regarding conductive hearing loss due to ossicular chain disruption [6].

Transverse fractures often affect the labyrinth, the vestibular, and cochlear systems, as well as the facial nerve. Longitudinal fractures (sparing the otic capsule) commonly involve the external auditory canal, tympanic membrane, and the middle ear including the ossicular chain, which often results in conductive hearing loss [5, 7, 8]. Immediate surgery is only indicated in patients with primary facial nerve palsy due to traumatic neurotmesis or in patients with labyrinthine fistula (e.g. rupture of round window membrane) leading to perilymph loss and deafness [1, 9]. Cerebrospinal fluid leaks might also need early surgical intervention if conservative treatment fails. Spontaneous recovery of traumatic conductive hearing loss is reported in 77% with conservative treatment [4]. Reconstructive middle ear surgery is indicated as a second stage elective procedure in patients with persistent conductive hearing loss and/or traumatic rupture of the tympanic membrane [1, 2]. The prognosis for hearing after middle ear surgery is excellent [2, 10–12].

We sought to perform a retrospective analysis of patients with a radiologically suspected ossicular chain dislocation and to assess the radiologic CT parameters and their association with conductive hearing outcome.

## Material and methods

### Patient population

This retrospective single-center cohort study included patients admitted to the emergency department (ED) with an ossicular injury caused by trauma and a fully documented audiometric examination at the time of the traumatic event in the period January 2010 to December 2017. All patients underwent a CT scan of the head as part of our standard emergency procedure for trauma patients.

In a primary screening of the radiological database, head trauma cases including petrous bone injury were identified using key words like "ossicular dislocation" "ossicular dehiscence" "petrous bone" by an experienced head and neck neuroradiologist (FW). In the second-stage screening conducted by a medical student (NS), only patients with a reported or suspected traumatic ossicular chain dislocation were included. All images were reviewed and assessed by 2 blinded neuroradiologists; one was a very experienced head and neck neuroradiologist (FW) and the other was a neuroradiology trainee (MK).

We excluded patients whose scans showed no proven dislocation of the ossicular chain in CT or had no head trauma. We further excluded patients with bilateral temporal bone fracture since the contralateral healthy side served as its own control. S1 Fig in S1 Appendix shows how patients were selected for the analysis.

## Radiological assessment: Parameters

All CT examinations were performed with the patient in a supine position using a 128-slice CT scanner (SOMATOM® Definition Edge; Siemens Healthcare, Erlangen, Germany). A certified reporting workstation (Sectra IDS7, Linköping, Sweden) was used for evaluation by the 2 neuroradiologists, who were blinded to outcomes. Slight motion artifacts were considered acceptable. All the images collected were of sufficiently good quality to allow an accurate assessment of the middle ear.

Overall, 4002 CT scans of trauma patients were retrospectively reviewed and, of these, 34 patients with ossicular joint dislocation due to trauma were included in the study.

Image reconstruction according to our standard in-house trauma protocol included a soft-tissue window (kernel J45s) and a bone window (kernel J70h) of the acquired CT scan of the head; each in the axial, coronal, and sagittal plane. For our retrospective data analysis, we additionally reconstructed the CT scans in 3-D to provide a different view of the ossicular chain anomalies in our trauma cohort. Both, 2-D and 3-D views have been used for the measurements [13].

The acquisition parameters of our standard trauma CT scan of the head were: slice thickness 1.0 mm, matrix $512 \times 512$, field of view 200 mm, total acquisition time of 1 second by tube current-time product of 240 mA, and tube voltage 100 kV. This resulted on average in a computed tomography dose index of 35 mGy and a dose-length product of 650 mGycm.

Assessment of the presence of an ossicle fracture was followed by the evaluation of ossicle dislocation or luxation based on visual identification of a discernable gap between ossicles. The distance 'D' between malleus and incus was measured by drawing a virtual line between the long axis of the incus and the middle of the head of the malleus, and the offset of the incus in medial or lateral deviation measured in mm were reported (Fig 1). The presence or absence of luxation and/or dislocation of the ossicles was recorded as: none, incudostapedial, incudo-malleolar, stapedo-vestibular, or complex if there was a luxation or dislocation in more than one direction (dislocation of several axis).

The degree of the offset between malleus and incus dislocation was separately measured, and the malleus–incus axis angle 'α' measured at midpoints (degrees) and the malleus–incus axis angle 'β' (degrees) were documented (Fig 1). Furthermore, we measured the ossicular joint space 'd' in mm in the soft tissue and bone window to determine whether the appearance of the space was normal or was filled with hemorrhage or air, based on standard Hounsfield Units. Fig 1 shows the definitions of all continuous variables (measurement of distances and angles).

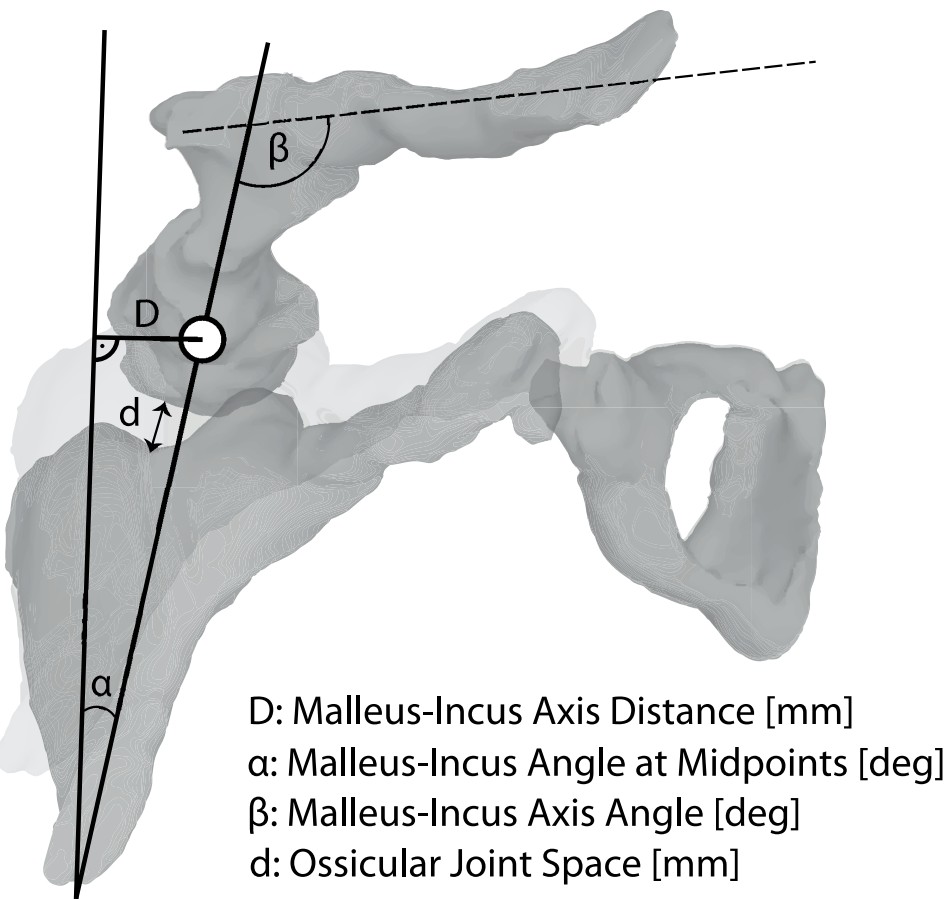

D: Malleus-Incus Axis Distance [mm]
α: Malleus-Incus Angle at Midpoints [deg]
β: Malleus-Incus Axis Angle [deg]
d: Ossicular Joint Space [mm]

**Fig 1. Radiologic measurement parameters.** Definition of all continuous variables and radiologically measured parameters.

## Audiometric testing

We assessed hearing outcome 1–3 months after the index ED visit and after resorption of any hemotympanon confirmed by tympanometry and otoscopy. Mixed hearing loss was not an exclusion criterion, however, sensorineural hearing loss due to labyrinthine concussion was not assessed since it was not considered a primary or secondary endpoint in this study. We used data from pure tone audiometry with air and bone conduction. The pure tone average including frequencies of 500Hz, 1000Hz, 2000Hz, and 3000Hz was calculated following the guidelines of the American Academy of Otolaryngology Committee on Hearing and Equilibrium [14]. An average air-bone gap <20 dB was considered a good hearing outcome.

## Statistics

We used SPSS (IBM Version 25) for statistical analysis. We calculated the inter-rater agreement for each variable, Cohen's kappa for categorical variables and the intraclass correlation coefficient for continuous data. A logistic regression was applied for the assessment of independent variables that might determine a poor hearing outcome with an air-bone gap of ≥20dB. Hearing outcome was coded as a binary, dependent variable.

A receiver operator characteristics (ROC) curve was constructed using the radiological variables. We assessed the diagnostic accuracy and calculated the optimal cut-point (Youden's index) for the discrimination of poor hearing outcome. Normative data from the contralateral healthy ear (2 standard deviations from the mean) and data from the ROC curve served as a basis for the classification of traumatic ossicular chain luxations.

## Ethics

The institutional review board and the local ethics committee (Kantonale Ethikkommission des Kantons Bern, Schweiz) gave approval for the access to and use of the data collected with the intention of using it for retrospective clinical research. Data have been fully anonymized. Informed consent was waived by the ethics committee.

## Results

We included 34 patients (12 females) with a mean age of 44.7 years (SD 24.9), 20 with a dislocation of the incudomalleolar joint, 1 with involvement of the incudostapedial joint, 4 with a joint distension (malleus head still within the facet of the incus without axis deviation but expanded joint space), and 9 with a complex dislocation involving all 3 ossicles. All patients had a history of head trauma, 56% reported a fall from height, 27% were involved in a traffic accident, 11% had a blunt head trauma and 6% had an unknown trauma mechanism. Fifteen of the 34 patients had a severe brain injury with a Glasgow Coma Scale (GCS) score of 3–8, 5 patients had a moderate (GCS 9–12), and 6 patients a minor brain injury (GCS 13–15). The state of consciousness remained unknown for 8 patients. The main complaints at the index ED visit were dizziness (6 out of 34) and hearing loss (7) followed by nausea (3), vomiting (3), headache (3), facial palsy (2), otorrhea (2), and gait disturbance (1).

One-third of the patients had facial nerve palsy (12 out of 34). Twenty-two patients suffered from bloody ear discharge, 10 had a hemotympanon, and 1 patient a perforation of the tympanic membrane; however, only 7 patients complained of hearing loss. Nine of the 22 patients who underwent follow-up hearing tests 1–3 months after the initial trauma had a hearing loss ≥20dB (S1 Fig in S1 Appendix).

Radiological examination revealed that 25 patients had a longitudinal, 1 patient a transverse, 7 patients a mixed/complex temporal bone fracture. One trauma patient had no temporal bone fracture, another patient showed a fracture of the incus.

Inter-rater agreement on the radiological parameters for the assessment of ossicular chain luxations was excellent (see S1 Table in S1 Appendix), except regarding the categorical assessment of the incus axis deviation. Normative data for the incudomalleolar joint are shown in the Appendix (S2 Table in S1 Appendix).

The distance 'D' [mm] measured between the 2 axes from malleus and incus (Fig 1) was the most significant factor for predicting poor hearing outcome (air bone gap ≥20dB). The probability of poor hearing outcome increased if the measured deviation of incus and malleus axis increased (odds ratio (OR) 2.67 per 0.1mm, 95% CI, 1.32–5.41, $p$ = .006, Table 1).

The angle 'α' and the distance 'D' between the 2 measured axes also had the highest sensitivity and specificity for predicting a poor hearing outcome (Table 2). A cut-off axis distance of 0.25mm showed a sensitivity of 0.778 and a specificity of 0.94 ($p$ < .001) for discrimination between poor and good hearing outcome in terms of conductive hearing loss. The axis angle measured at the midpoints 'α' was also a significant predictor (OR 1.78, 95% CI, 1.05–3.00, $p$ = .03) with a high sensitivity (0.89, $p$ = .002) but lower specificity (0.743).

Fig 2 shows a receiver operator characteristics (ROC) curve with all the applied radiological parameters.

**Table 1. Logistic regression for hearing outcome ≥20dB.**

| Parameter | Odds Ratio | Lower 95% CI | Upper 95% CI | *P* Value |
|---|---|---|---|---|
| Malleus-incus axis distance 'D' (per 1/10mm increase) | **2.674** | 1.32 | 5.416 | **.006** |
| Malleus-incus axis angle measured at midpoints 'α' (per degree increase) | **1.781** | 1.056 | 3.003 | **.03** |
| Malleus-incus axis angle 'β' (per degree increase) | 1.033 | 0.979 | 1.09 | .232 |
| Ossicle joint space 'd' (per 1/10mm increase) | 1.374 | 0.948 | 1.991 | .094 |

**Table 2. ROC (receiver operator characteristics) curves.**

| Parameter | Cut-off | Sensitivity | Specificity | AUC | *P* Value |
|---|---|---|---|---|---|
| Malleus-incus axis distance 'D' (mm) | **0.250** | 0.778 | 0.943 | 0.892 | < **.001** |
| Malleus-incus angle measured at midpoints 'α' (deg) | **0.950** | 0.889 | 0.743 | 0.840 | **.002** |
| Malleus-incus axis angle 'β' (deg) | 93.8 | 0.222 | 1.000 | 0.524 | .827 |
| Ossicle joint space 'd'(mm) | 0.850 | 0.500 | 0.829 | 0.664 | .203 |

Abbreviations: AUC, area under the curve.

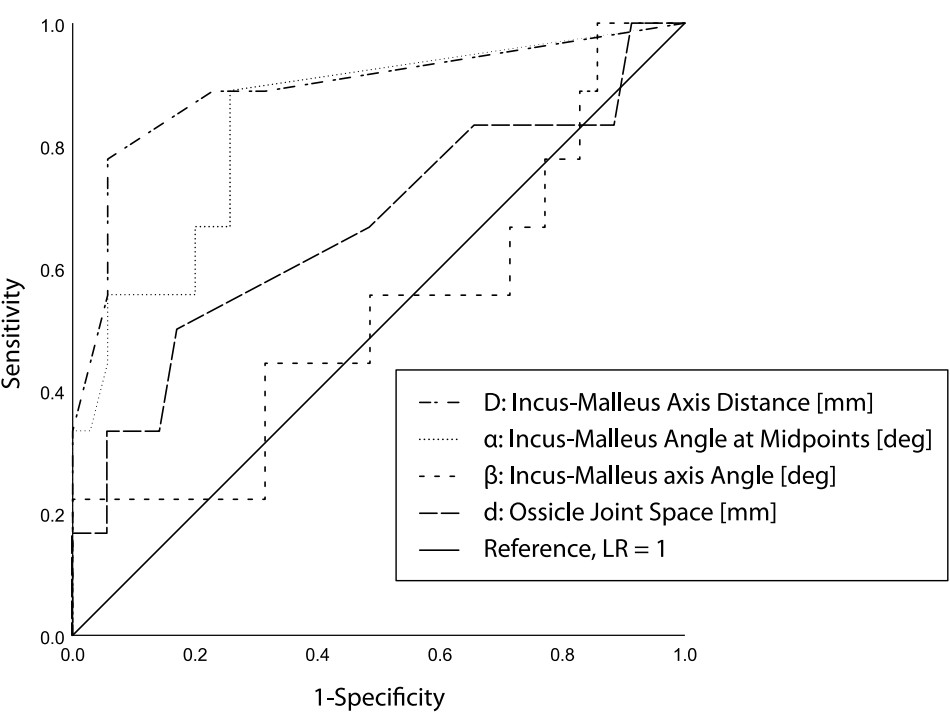

**Fig 2. Receiver operator characteristics (ROC) curve.** ROC curve for all 4 radiological parameters. The best predictive parameter was the distance between the 2 axes [mm], which yielded a curve bending toward the left top corner. Curves along the diagonal line, such as the parameter "malleus-incus axis angle 'β'" reflect a random guess and are not discriminative with respect to hearing outcome.

**Table 3. New classification system of ossicular chain dislocation.**

| Grade | Malleus-incus axis distance 'D' (mm) | Dislocation/subluxation | Hearing outcome[a] | Example |
|---|---|---|---|---|
| I | 0–0.07 | No | Normal | Fig 2A |
| II | 0.08–0.25 | Yes | Normal | Fig 2B |
| III | >0.25 | Yes | Poor[b] | Fig 2C |

[a]1-3 months after trauma

[b]PTA air bone gap >20dB HL

Table 3 shows our new 3-grade classification for the prediction of normal or poor hearing outcome based on the distance between incus and malleus axis. Grade I represents the normal, non-displaced axis distance of the incudomalleolar joint (Fig 3A, normative Data). Grade II includes cases with an anatomical axis configuration in the pathological range, but whose predicted hearing outcome is good (Fig 3B). Patients with grade III axis distance have a poorer prognosis for hearing outcome due to the abnormal incudomalleolar joint dislocation (Fig 3C). The discrimination cut-off between Grade II and Grade III was derived from the ROC analysis (Table 2).

## Discussion

Almost half of the patients suffered from conductive hearing loss 1–3 months after temporal bone fracture with associated luxation/dislocation of middle ear ossicles. High resolution temporal bone CTs with narrow slice widths allowed accurate radiologic measurements with high inter-rater agreement. The distance 'D' between the 2 axes through the short process of the incus and the midpoint of the malleus was an accurate predictor for hearing outcome. Each deviation difference of 1/10mm increased the odds for a poor predicted hearing outcome by 2.6. A cut-off value of 0.25mm had a statistically significant high sensitivity and specificity for discriminating poor from normal hearing outcome and served as the threshold for a new proposed classification.

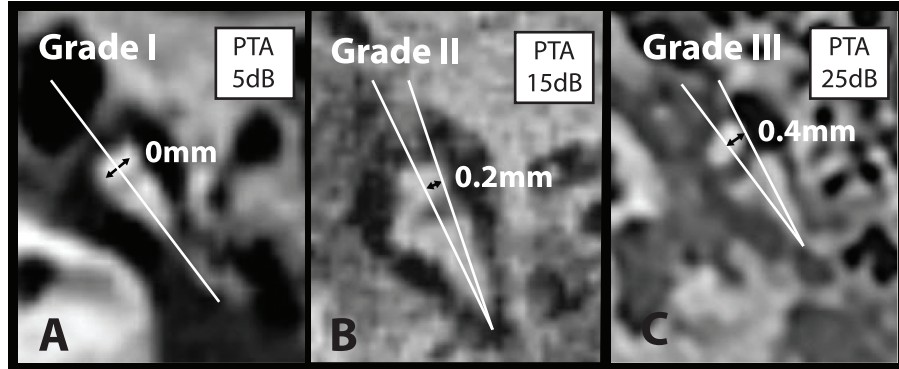

**Fig 3. Computed tomography examples of malleus-incus axis distance 'D'.** Panel A, B and C shows a computed tomography (at index visit) of the incudomalleolar joint from 3 patients and the distances 'D' (mm) between the malleus axis and incus axis for grade I, II and III and the corresponding hearing outcome (pure tone average (PTA) of the air bone gap), dB HL) 1–3 months after trauma. The image in panel C was horizontally flipped for better visualization.

The most frequent trauma mechanism was falls from height followed by traffic accidents. A rapid deceleration or high energetic trauma was the cause of temporal bone fractures with consecutive injury to the ossicular chain. This result is congruent with that of a recent study by Delrue et al. [11], but in most studies the main cause of injury was a traffic accident [2, 4, 7, 10].

There was a high proportion of longitudinal fractures in our cohort. The classical classification of temporal bone fractures into longitudinal, transverse, or mixed fractures, however, has a poor correlation with clinical symptoms [15, 16]. Therefore, new classifications have been proposed taking into account the involvement of the otic capsule [15], the petrous bone [17], or 4 parts of the temporal bone (squama, tympanic, mastoid, and petrous) [18]. These studies showed that only the categorization into petrous or non-petrous fractures was significantly associated with the clinical symptoms [17–19]. Since we only included patients with petrous bone involvement, we decided to use the traditional classification, according to which the exact description of the fracture course and the affected regions is essential.

Ossicular chain dislocation or luxation is a frequent complication of temporal bone fracture. In this study, incudomalleolar dislocation was by far the most common type, as has been reported in other CT-based studies [20, 21]. However, surgical explorations or cadaveric dissections revealed frequent involvement of the incudostapedial joint [7, 8, 10, 22, 23]. This is plausible because dislocation of the incus, which is embedded between the malleus and stapes, should affect both joints. Another plausible explanation for the lower prevalence of the often subtle incudostapedial dislocation in our CT-based study might be the initial hemotympanum, which makes an accurate radiologic assessment difficult, especially in early trauma diagnostics and for slight dislocations [20, 24].

Our study data suggest that a latero-medial dislocation of the incus is more likely than a longitudinal traction. A lateral displacement changes the axis angle and axis distance, whereas traction leads to a widening of the joint space. The configuration of the ligamentous apparatus of the middle ear ossicles supports the observed mechanism of dislocation and the vulnerability of the incus. While the malleus is attached by 3 ligament folds (anterior, lateral, and posterior), the incus is only kept in place by a strong posterior incudal ligament fold, which is attached at the apex of the short incus process [25] (S2 Fig in S1 Appendix). The lateral incudomalleal fold, however, is purely membranous and is probably not sufficient to protect against shearing forces. In addition, the stapes and malleus are attached to the middle ear muscles, the tensor tympani muscle, and the stapedial muscle, resulting in stronger stabilization of these 2 ossicles. A dislocation of the malleus is less likely, since it is additionally attached to the fibrous layers of the tympanic membrane [7, 26, 27]. A large proportion of dislocated incudomalleolar joints showed a pattern of subluxation, but nevertheless led to a satisfactory hearing outcome in the long term. Reversible subluxations may only stretch the ligamentous apparatus. Possible spontaneous resolution of ossicular chain dislocations has been described by other authors [4, 28]. Such luxations were classified as grade II (Table 3).

## Strengths and limitations

To our knowledge, this is the first study to propose a grading system for predicting conductive hearing outcome based on CT in patients with traumatic dislocations of the middle ear ossicles. This classification of hearing outcome prediction is limited to patients with conductive or mixed hearing loss after trauma. Sensorineural hearing loss due to labyrinthine concussion is not covered; however, sensation levels might be screened by the Weber tuning fork test at the bedside (lateralization toward the affected middle ear or the healthy inner ear) or assessed by bone conduction hearing tests. Patients classified as grade II had a pathological axis deviation

in the index CT but still had normal hearing after 1–3 months; however, no follow-up imaging or surgical reports were available to confirm recovery in terms of anatomy.

CT spatial resolution was limited to 0.2–0.3mm, however, distance measurements close to the resolution limits were not performed between anatomical structures but were restricted to distances between two marked axis. Some calculated mean distances in our classification table were smaller than the spatial CT resolution which is the result from statistical analysis including values with zero millimeters (no dislocation). Further studies using the proposed classification system and its radiologic measuring techniques are necessary to detect possible technical limitations regarding the measurements. Overall, the inter-rater agreement was excellent indicating a reproducible and accurate radiologic measurement, which was also found by Maillot et al, provided that assessments were done by senior readers and taking into account 3D CT reconstructions [13].

Finally, luxations of the incudostapedial joint were not measured quantitatively due to the low frequency of occurrence in our cohort; however, qualitative radiological assessment was still possible. It is unclear whether such isolated joint luxations remain limited to the incudostapedial joint or—more likely—affect the entire ossicular chain.

## Potential implications

This proposal for a grading system is a first attempt to classify patients according to whether a normal or poor hearing outcome is likely after temporal bone fractures. Future prospective studies are essential for its validation. Prospective studies with hearing assessments after resorption of the hemotympanon, long-term hearing results and intraoperative findings might give more insights about the mechanism and morphology of dislocations. Information about prognosis and encouragement might improve the follow-up of these patients.

## Conclusions

Assessment of high-resolution CT scans of temporal bone with respect to ossicular chain dislocations after traumatic temporal bone fractures was feasible. Axis distance 'D' of the short incus process and the middle point of the malleus body were strongly predictive for hearing outcome in terms of air conduction 1–3 months after trauma. We propose a new 3-level classification system for the prediction of poor or normal hearing outcome in patients with ossicular chain dislocation caused by trauma based on radiologic measures.

## Supporting information

**S1 Appendix.**
(PDF)

## Author Contributions

**Conceptualization:** Georgios Mantokoudis, Franca Wagner.

**Data curation:** Njima Schläpfer, Manuel Kellinghaus, Arsany Hakim, Moritz von Werdt, Franca Wagner.

**Formal analysis:** Georgios Mantokoudis.

**Investigation:** Njima Schläpfer.

**Methodology:** Georgios Mantokoudis, Franca Wagner.

**Project administration:** Georgios Mantokoudis, Marco D. Caversaccio.

**Resources:** Georgios Mantokoudis, Marco D. Caversaccio, Franca Wagner.

**Supervision:** Georgios Mantokoudis, Marco D. Caversaccio, Franca Wagner.

**Validation:** Georgios Mantokoudis, Manuel Kellinghaus, Arsany Hakim, Franca Wagner.

**Writing – original draft:** Georgios Mantokoudis, Njima Schläpfer, Franca Wagner.

**Writing – review & editing:** Georgios Mantokoudis, Njima Schläpfer, Manuel Kellinghaus, Arsany Hakim, Moritz von Werdt, Marco D. Caversaccio, Franca Wagner.

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
