## [Decision Letter · Decision Letter 0]

13 Oct 2020

PONE-D-20-27974

Traumatic Dislocation of Middle Ear Ossicles: A New Computed Tomography Classification Predicting Hearing Outcome

PLOS ONE

Dear Dr. Mantokoudis,

Thank you for submitting your manuscript to PLOS ONE. After careful consideration, we feel that it has merit but does not fully meet PLOS ONE’s publication criteria as it currently stands. Therefore, we invite you to submit a revised version of the manuscript that addresses the points raised during the review process.

We look forward to receiving your revised manuscript.

Kind regards,

Rafael da Costa Monsanto, M.D.

Academic Editor

PLOS ONE

Additional Editor Comments:

Although the reviewers considered the study well-written and scientifically sound, there are still some issues that need to be addressed before the manuscript is further considered for publications. The most critical concerns were regarding description of the methodology and results, especially concerning to patient selection and analysis of the results.

Journal Requirements:

2.  Thank you for including your ethics statement:  "The institutional review board and the local ethics committee (KEK: EK BE 2018-00062) gave approval for the access to and use of the data collected with the intention of using it for retrospective clinical research.".   

3. Please provide additional details regarding participant consent. In the ethics statement in the Methods and online submission information, please ensure that you have specified (i) whether consent was informed and (ii) what type you obtained (for instance, written or verbal, and if verbal, how it was documented and witnessed). If your study included minors, state whether you obtained consent from parents or guardians. If the need for consent was waived by the ethics committee, please include this information.

Reviewers' comments:

Reviewer's Responses to Questions

**Comments to the Author**

1. Is the manuscript technically sound, and do the data support the conclusions?

Reviewer #1: Yes

Reviewer #2: Yes

Reviewer #3: Yes

2. Has the statistical analysis been performed appropriately and rigorously? 

Reviewer #1: Yes

Reviewer #2: Yes

Reviewer #3: Yes

3. Have the authors made all data underlying the findings in their manuscript fully available?

Reviewer #1: Yes

Reviewer #2: Yes

Reviewer #3: Yes

4. Is the manuscript presented in an intelligible fashion and written in standard English?

Reviewer #1: Yes

Reviewer #2: Yes

Reviewer #3: Yes

5. Review Comments to the Author

Reviewer #1: I congratulate the authors for this interesting paper.

How unfortunate that the study managed to analyze only 34 exams in a universe of more than 4000 CT scans. Anyway, the authors managed to adequately point out the limitations of the study and I believe that the idea should follow so that other studies use the same method to validate the tomographic correlation with the potential to predict the audiometric results.

Reviewer #2: This is a review of the temporal bone CT scans of patients with a radiologically suspectedossicular chain dislocation and their association with conductive hearing outcome. It states that Figure S1 in the Appendix shows how patients were included in the analysis, although I failed to find this figure in my copy of the manuscript. The severity of incudomalleal joint disruption was assessed with 1 mm slices of standard trauma CT scan of the head. Aside from the inter rater agreement, logistic regression was used to assess the association between 4 CT scan parameters with poor hearing outcome defined as conductive hearing loss of ≥20dB air-bone gap. The authors then found that the malleus-incus axis distance and malleus-incus angle measured at midpoints were statistically correlated with conductive hearing loss.

Here are some of my questions:

1. Was the patient selection process blinded to outcomes? Who did the selection, how was this done and how were disagreements to include patients addressed?

2. Were the 2 neuroradialogists blinded to outcomes?

3. I am concerned that the distance of 0.25 mm was found significant when the cuts were 1 mm in thickness. The “ice cream cone” constituting the incus and the malleal head appears discernible enough from the sample CT scan picture but how were differences in measurements settled? And how high were the inter rater agreements in each of the 4 radiologic parameters? (I did not find Appendix Table S1 in my copy).

4. I am not clear as to how the authors determined the ranges of the malleus-incus axis distances of 0-0.07, 0.08-0.25 and >0.25 for each category. I may have missed how they have correlated these ranges with normal or abnormal hearing.

I appreciate how the discussion attempts to explain the results in terms of the ligamentous attachments of the malleus and incus may explain the vulnerability of the incudomalleal joint to trauma. An illustration may more clearly drive home the point.

I also appreciate the limitations part and agree with all the points raised by the authors, particularly with their statement that “some calculated mean distances in our classification table were smaller than the spatial CT resolution which is the result from statistical analysis”. Given this potential technical difficulty in measurement, I think they should point out that their new classification system should be applied into another set of trauma patients to understand how well it performs.

Reviewer #3: The authors perform a study to determine the association between CT findings of post-traumatic ossicular injuries and conductive hearing outcome. They also propose a classification system to predict audiometric outcomes based on radiological measurements. Although this is an interesting area of research to explore, there are some major concerns that need to be addressed, as listed below:

1. The abstract should be revised to include more detail about methodology, such as summary of inclusion/exclusion criteria, measurements taken (e.g. incus-malleus axis distance, etc) and how good/poor hearing was defined by the authors. “CT” abbreviation should be provided in the Objectives.

2. Also, in the abstract — The authors write in the conclusion: “Adequate assessment of high resolution CT scans of temporal bone in which ossicular chain dislocation had occurred after traumatic fractures of temporal bone was feasible.” — did you mean your new radiological parameters (measurements) for assessment were feasible and reproducible? If the authors also wanted to assess their feasibility, it should be included in the aims (objectives).

3. Lines 80-83: Patients should also acknowledge the fact that hearing dysfunction is often overlooked in polytrauma patients because other trauma-related physical/brain injuries that take medical priority.

4. Lines 84: fractures are OFTEN associated with a hemotympanum.

5. Material and Methods: Inclusion criteria is confusing — were considered cases with ossicular chain injury following head trauma regardless of presence of TBF? Or was presence of TBF determinant for study inclusion? I also suggest providing # of excluded cases within each criterion in Figure 1S (no dislocation of the ossicular chain, no temporal fracture after second review, or who had a bilateral temporal bone fracture).

6. Line 152: What does “more than one direction” mean? More than one joint? Please clarify.

7. Did any patient have a history of asymmetric hearing loss or ear disease prior to the trauma?

8. Line 208, 217 — Keep consistency in terminology while reporting hearing loss (average air-bone gap vs. conductive hearing loss) throughout the Results.

9. Audiometric data timeline considered for inclusion in the study needs clarification. Did all patients have audiometric evaluation? Did the authors include audiometric data before 1 mo in addition to those performed 1-3 months after trauma? In the Results, Line 207-208: “Nine of the 22 patients who underwent follow-up hearing tests still had a hearing loss ≥20dB 1-3 months after the initial trauma” gives the impression that audiometric data before 1mo were also reviewed. Please clarify.

10. Line 109: Figure 1 does not relate to the sentence/paragraph.

11. Line 164-173: It is not clear if cases with mixed hearing loss were also included. Please specify.

12. Line 197: “4 with a joint distension” — Please clarify.

13. Line 210-211: The total accounts for 33 cases. However, authors report a total of 34 cases in this study. Similar to my question #5 — were cases without TBF included? Please clarify.

14. Keep consistency when reporting measurements in the main manuscript, table and figures (e.g. Incus-malleus or malleus-incus). Also, consider including symbols (D, d, α and β) in the tables.

15. Line 225: Table 2 indicates that D and α had the highest sensitivity and specificity for predicting a poor hearing outcome instead. Please review/clarify.

16. In Methods, authors mention to have performed 3D-CT reconstructions to provide different views of the ossicular chain anomalies — Were both 2D-CT axial and sagittal views and 3D-CT reconstructions used for measurements? It should be clarified in the Methods. Also, were there any ossicle fractures identified in your cohort?

17. I suggest including the recent publication from Maillot et al. (2020) — PMID: 28551022 — related to this topic to enrich the discussion.

6. PLOS authors have the option to publish the peer review history of their article (what does this mean?). If published, this will include your full peer review and any attached files.

Reviewer #1: No

Reviewer #2: **Yes: **Jose Acuin

Reviewer #3: No

---

## [Author Response · Author response to Decision Letter 0]

26 Nov 2020

Reviewer #1: 

I congratulate the authors for this interesting paper.

How unfortunate that the study managed to analyze only 34 exams in a universe of more than 4000 CT scans. Anyway, the authors managed to adequately point out the limitations of the study and I believe that the idea should follow so that other studies use the same method to validate the tomographic correlation with the potential to predict the audiometric results.

AUTHOR REPLY: We would like to thank the reviewer for the thorough review of the manuscript, the positive feedback and the appreciation of the work.

Reviewer #2: 

This is a review of the temporal bone CT scans of patients with a radiologically suspected ossicular chain dislocation and their association with conductive hearing outcome. It states that Figure S1 in the Appendix shows how patients were included in the analysis, although I failed to find this figure in my copy of the manuscript. The severity of incudomalleal joint disruption was assessed with 1 mm slices of standard trauma CT scan of the head. Aside from the inter rater agreement, logistic regression was used to assess the association between 4 CT scan parameters with poor hearing outcome defined as conductive hearing loss of ≥20dB air-bone gap. The authors then found that the malleus-incus axis distance and malleus-incus angle measured at midpoints were statistically correlated with conductive hearing loss.

Here are some of my questions:

1. Was the patient selection process blinded to outcomes? Who did the selection, how was this done and how were disagreements to include patients addressed?

AUTHOR REPLY: Thank you for raising this important question. In a primary screening the radiological database has been searched for key words like “ossicular dislocation” “ossicular dehiscence” “petrous bone” by a experienced head and neck neuroradiologist (FW). In the second-stage screening, all reports of the 4002 CT scans were screened by a medical student (NS) for reported or suspected ossicular chain dislocation associated with trauma. All images were then again reviewed and assessed by 2 blinded neuroradiologists to determine whether a dislocation actually was present or not. Neurologist were blinded to outcomes. We have added this specification to “Material and Methods: Patient Population.

2. Were the 2 neuroradialogists blinded to outcomes?

AUTHOR REPLY: Both neuroradiologists were blinded to outcomes as stated in Material and Methods under the section “Patient population”.

REVISED: “A certified reporting workstation (Sectra IDS7, Linköping, Sweden) was used for evaluation by the 2 neuroradiologists, who were blinded to outcomes.”

3. I am concerned that the distance of 0.25 mm was found significant when the cuts were 1 mm in thickness. The “ice cream cone” constituting the incus and the malleal head appears discernible enough from the sample CT scan picture but how were differences in measurements settled? And how high were the inter rater agreements in each of the 4 radiologic parameters? (I did not find Appendix Table S1 in my copy).

AUTHOR REPLY: Please see appendix table 1S attached

REVISED: Limitations section: Some calculated mean distances in our classification table were smaller than the spatial CT resolution which is the result from statistical analysis including values with zero millimeters (no dislocation).

4. I am not clear as to how the authors determined the ranges of the malleus-incus axis distances of 0-0.07, 0.08-0.25 and >0.25 for each category. I may have missed how they have correlated these ranges with normal or abnormal hearing.¨

AUTHOR REPLY: Grade I was determined by the mean range of the contralesional, healthy side (normative data, mean + two standard deviations = 0.07). For the determination of the cut-off between grad 2 and 3, we used the ROC analysis and calculated the best discrimination cut-off (0.25). We added this information in the manuscript to make it more clear.

REVISED: “Grade I represents the normal, non-displaced axis distance of the incudomalleolar joint (Figure 3A, normative Data).”

“The discrimination cut-off between Grade II and Grade III was derived from the ROC analysis (Table 2).”

I appreciate how the discussion attempts to explain the results in terms of the ligamentous attachments of the malleus and incus may explain the vulnerability of the incudomalleal joint to trauma. An illustration may more clearly drive home the point.

AUTHOR REPLY: We added an additional figure in the appendix illustrating the incudal ligament folds (figure S2).

I also appreciate the limitations part and agree with all the points raised by the authors, particularly with their statement that “some calculated mean distances in our classification table were smaller than the spatial CT resolution which is the result from statistical analysis”. Given this potential technical difficulty in measurement, I think they should point out that their new classification system should be applied into another set of trauma patients to understand how well it performs.

AUTHOR REPLY: We agree with the reviewer that the new classification should be applied to other trauma patients.

REVISED: “Further studies using the proposed classification system and its radiologic measuring techniques are necessary to detect possible technical limitations regarding the measurements.”

Reviewer #3: 

The authors perform a study to determine the association between CT findings of post-traumatic ossicular injuries and conductive hearing outcome. They also propose a classification system to predict audiometric outcomes based on radiological measurements. Although this is an interesting area of research to explore, there are some major concerns that need to be addressed, as listed below:

1. The abstract should be revised to include more detail about methodology, such as summary of inclusion/exclusion criteria, measurements taken (e.g. incus-malleus axis distance, etc) and how good/poor hearing was defined by the authors. “CT” abbreviation should be provided in the Objectives.

AUTHOR REPLY: Thank you for raising these important points. We revised the abstract and added inclusion/exclusion criteria. “CT” abbreviation provided. Radiologic measurements and hearing outcome defined.

2. Also, in the abstract — The authors write in the conclusion: “Adequate assessment of high resolution CT scans of temporal bone in which ossicular chain dislocation had occurred after traumatic fractures of temporal bone was feasible.” — did you mean your new radiological parameters (measurements) for assessment were feasible and reproducible? If the authors also wanted to assess their feasibility, it should be included in the aims (objectives).

AUTHOR REPLY: We assessed also the feasibility of radiological measurements and added this to the objectives in the abstract.

REVISED: “To assess the feasibility of radiologic measurements and find out whether hearing outcome could be predicted based on computer tomography (CT) scan evaluation in patients with temporal bone fractures and suspected ossicular joint dislocation.”

3. Lines 80-83: Patients should also acknowledge the fact that hearing dysfunction is often overlooked in polytrauma patients because other trauma-related physical/brain injuries that take medical priority.

AUTHOR REPLY: Corrected. We integrated this important fact in the introduction.

REVISED: “Often hearing dysfunction is overlooked in polytrauma patients because other trauma-related physical/brain injuries take medical priority.”

4. Lines 84: fractures are OFTEN associated with a hemotympanum.

AUTHOR REPLY: corrected

5. Material and Methods: Inclusion criteria is confusing — were considered cases with ossicular chain injury following head trauma regardless of presence of TBF? Or was presence of TBF determinant for study inclusion? I also suggest providing # of excluded cases within each criterion in Figure 1S (no dislocation of the ossicular chain, no temporal fracture after second review, or who had a bilateral temporal bone fracture).

AUTHOR REPLY: We agree that exclusion criteria were not clearly stated. We rephrased this sentence. Yes, we included cases with ossicular chain injury following any head trauma regardless of presence of TBF. However, we did not found any bilateral temporal bone fractures with associated bilateral ossicular dislocation (n=0). We corrected Figure S1. 

REVISED: “We excluded patients whose scans showed no proven dislocation of the ossicular chain in CT or had no head trauma. We further excluded patients with bilateral temporal bone fracture since the contralateral healthy side served as its own control.”

6. Line 152: What does “more than one direction” mean? More than one joint? Please clarify.

AUTHOR REPLY: “more than one direction” means a dislocation in several planes (axes). 

REVISED: or complex if there was a luxation or dislocation in more than one direction (dislocation of several axis). 

7. Did any patient have a history of asymmetric hearing loss or ear disease prior to the trauma?

AUTHOR REPLY: These patients had no audiological assessment prior to the trauma, however, any preexisting asymmetric hearing cannot be excluded due to the retrospective nature of our study. 

8. Line 208, 217 — Keep consistency in terminology while reporting hearing loss (average air-bone gap vs. conductive hearing loss) throughout the Results.

AUTHOR REPLY: corrected

9. Audiometric data timeline considered for inclusion in the study needs clarification. Did all patients have audiometric evaluation? Did the authors include audiometric data before 1 mo in addition to those performed 1-3 months after trauma? In the Results, Line 207-208: “Nine of the 22 patients who underwent follow-up hearing tests still had a hearing loss ≥20dB 1-3 months after the initial trauma” gives the impression that audiometric data before 1mo were also reviewed. Please clarify.

AUTHOR REPLY: We did not review audiologic data in the time period from trauma until 1 month after trauma. We rephrased this sentence.

10. Line 109: Figure 1 does not relate to the sentence/paragraph.

AUTHOR REPLY: We agree. This relates to figure S1 in the appendix. 

11. Line 164-173: It is not clear if cases with mixed hearing loss were also included. Please specify.

AUTHOR REPLY: Thank you for pointing out this ambiguity. Mixed hearing loss was not an exclusion criterion. The study focuses on the ABG, whereas sensorineural hearing loss due to labyrinthine concussion was not assessed since it was not considered a primary or secondary endpoint in this study. We rephrased this in the methods section. We made it more clear in the methods section.

12. Line 197: “4 with a joint distension” — Please clarify.

AUTHOR REPLY: Joint distension means, that the malleus head was still within in the facet of the incus without axis deviation but the joint space was expanded. We revised accordingly.

13. Line 210-211: The total accounts for 33 cases. However, authors report a total of 34 cases in this study. Similar to my question #5 — were cases without TBF included? Please clarify.

AUTHOR REPLY: One trauma patient had no TBF. We added this in line 211 and clarified in the methods section. 

14. Keep consistency when reporting measurements in the main manuscript, tables and figures (e.g. Incus-malleus or malleus-incus). Also, consider including symbols (D, d, α and β) in the tables.

AUTHOR REPLY: Thank you for pointing out these inconsistencies. In the revised manuscript the term “malleus-incus” is now used consistently and we included the symbols (D, d, α and β) consistently throughout the whole manuscript, including tables, figures and the appendix.

15. Line 225: Table 2 indicates that D and α had the highest sensitivity and specificity for predicting a poor hearing outcome instead. Please review/clarify.

AUTHOR REPLY: Corrected. It should be ‘D’ and ‘α’». We added labels ‘D’, ‘α’, ‘β’ and ‘d’ to all tables.

16. In Methods, authors mention to have performed 3D-CT reconstructions to provide different views of the ossicular chain anomalies — Were both 2D-CT axial and sagittal views and 3D-CT reconstructions used for measurements? It should be clarified in the Methods. Also, were there any ossicle fractures identified in your cohort?

AUTHOR REPLY: We used both, 2D and 3D views for the measurements. We clarified this in the methods. One patient had a fracture of the incus. This was added in the results section. 

17. I suggest including the recent publication from Maillot et al. (2020) — PMID: 28551022 — related to this topic to enrich the discussion.

AUTHOR REPLY: We included the proposed publication in the methods and discussion.

---

## [Decision Letter · Decision Letter 1]

8 Jan 2021

Traumatic Dislocation of Middle Ear Ossicles: A New Computed Tomography Classification Predicting Hearing Outcome

PONE-D-20-27974R1

Dear Dr. Mantokoudis,

We’re pleased to inform you that your manuscript has been judged scientifically suitable for publication and will be formally accepted for publication once it meets all outstanding technical requirements.

Kind regards,

Rafael da Costa Monsanto, M.D.

Academic Editor

PLOS ONE

Additional Editor Comments (optional):

Congratulations on the excellent piece of work.

Reviewers' comments:

Reviewer's Responses to Questions

**Comments to the Author**

1. If the authors have adequately addressed your comments raised in a previous round of review and you feel that this manuscript is now acceptable for publication, you may indicate that here to bypass the “Comments to the Author” section, enter your conflict of interest statement in the “Confidential to Editor” section, and submit your "Accept" recommendation.

Reviewer #1: All comments have been addressed

Reviewer #3: All comments have been addressed

2. Is the manuscript technically sound, and do the data support the conclusions?

Reviewer #1: Yes

Reviewer #3: Yes

3. Has the statistical analysis been performed appropriately and rigorously? 

Reviewer #1: Yes

Reviewer #3: Yes

4. Have the authors made all data underlying the findings in their manuscript fully available?

Reviewer #1: Yes

Reviewer #3: Yes

5. Is the manuscript presented in an intelligible fashion and written in standard English?

Reviewer #1: Yes

Reviewer #3: Yes

6. Review Comments to the Author

Reviewer #1: I congratulate the authors to address a point-by-point revised document of their study. And I congratulate the authors for their very interesting work.

Reviewer #3: I would like to thank the authors for the revised manuscript. The paper is improved with the revisions made. I have no further comments or concerns.

7. PLOS authors have the option to publish the peer review history of their article (what does this mean?). If published, this will include your full peer review and any attached files.

Reviewer #1: No

Reviewer #3: No

---

## [Editor Report · Acceptance letter]

28 Jan 2021

PONE-D-20-27974R1 

Traumatic Dislocation of Middle Ear Ossicles: A New Computed Tomography Classification Predicting Hearing Outcome 

Dear Dr. Mantokoudis:

I'm pleased to inform you that your manuscript has been deemed suitable for publication in PLOS ONE. Congratulations! Your manuscript is now with our production department. 

Kind regards, 

on behalf of

Dr. Rafael da Costa Monsanto 

Academic Editor

PLOS ONE